# Managing conflicting goals through prioritization? The role of age and relative goal importance

**Alexandra M. Freund** [1,2,3] *, **Martin J. Tomasik** [1,2]

**1** Department of Psychology, University of Zurich, Zurich, Switzerland, **2** University Research Priority Program Dynamics of Healthy Aging, University of Zurich, Zurich, Switzerland, **3** National Centre of Competence in Research (NCCR) "LIVES – Overcoming Vulnerability: Life Course Perspectives"

* freund@psychologie.uzh.ch

**Data Availability Statement:** The data are available on OSF: at DOI 10.17605/OSF.IO/6PKBV.

**Funding:** Swiss National Science Foundation Grant 100014-138045 "Readiness to Disengage from Motivational Conflict: Adult Age Differences and

## Abstract

Three studies tested the role of prioritization in solving conflict between multiple goals in different age groups. Study 1 ($N$ = 185 young, middle-aged, older adults) stressed the importance to solve two competing tasks equally well within a short time. Older adults prioritized more than younger adults. However, contrary to our expectations, prioritization led to higher perceived conflict, more negative affect, and less control. Study 2 ($N$ = 117 younger and older adults) found that, using a more lenient instruction, deemphasizing the importance of performing equally well on both tasks, prioritization was no longer associated with perceived goal conflict. Study 3 ($N$ = 721 young, middle-aged, older adults) was an online study using hypothetical scenarios. This study was run to substantiate the potential mechanism underlying the differences between Study 1 and 2 and supported the hypothesized effect of the instructional strictness of pursuing two goals. Thus, when encountering conflicting goals older adults prioritize more than younger adults, but prioritization might not be optimal for solving short-term goal conflict when both conflicting goals are equally important.

## Introduction

Most people pursue multiple goals at the same time, some pertaining to different life domains such as work, family, or leisure, some to the same life domains [1]. Unfortunately, it is not easy to balance the demands of multiple goals that often draw on the same resources such as time, energy, or money, leading to goal conflict [2]. Most of the research on managing to pursue multiple goals has focused on the conflict between work- and family-related goals. This literature shows that goal conflict is typically related to negative outcomes such as lower goal attainment [3] and reduced health and well-being [4, 5]. A meta-analysis by Gray and colleagues [6] revealed that goal conflict is also related to lower positive psychological outcomes such as subjective well-being, but less strongly than to measures of psychological distress.

How, then, do people manage to pursue multiple goals without compromising their well-being when their goals vie for the same resources, thus being in conflict with each other? One process that allows people to pursue multiple goals without experiencing conflict is goal shielding. Goal shielding denotes the effect that the activation of one goal inhibits the cognitive availability of a competing goal [7]. This process allows people to temporarily prioritize one goal

Implications for Emotional Experiences" (PI: Alexandra M. Freund).

**Competing interests:** The authors have declared that no competing interests exist.

over another. This is in line with theorizing by Orehek and Vazeou-Nieuwenhuis [8] who identified *prioritizing* concurrent goals as a self-regulatory strategy that facilitates the management of multiple goals. However, Ballard and colleagues [9] showed that in the case of two conflicting goals, people generally tend to try to achieve both goals rather than clearly prioritize one of them, even if an extrinsic financial incentive would reward them for focusing on the achievement of only one of the goals. Ballard and colleagues [9] interpret this finding as indicating that people attach a psychological value to achieving a given goal (in their case a computer task that is in and off itself meaningless and unchallenging) that motivates them to pursue goals even if this strategy does not lead to maximizing extrinsic monetary rewards. Thus, prioritization of one goal over the other does not seem psychologically easy when facing a goal conflict.

It might also be helpful to consult the aging literature to address the question of how people manage goal conflict as studies have shown that older adults typically experience less conflict when asked to rate the relations between their personal goals [10, 11]. In line with the above-mentioned literature stressing the role of prioritization, one of the mechanisms underlying this age-related difference could be that older adults, due to the increasing limitation of resources based on physical and cognitive declines [12], are forced more often to prioritize one goal over the other when faced with goal conflicts, thereby focusing their more limited resources [13]. In fact, the literature on aging demonstrates that older adults prioritize more than younger adults by focusing their personal goals on personally important, superordinate goal domains [14]. As has been argued in the context of the model of selection, optimization, and compensation (SOC-model) [15], older adults are generally more selective because of the increasing limitation of resources, including a more limited future time perspective [16]. According to the SOC-model, prioritizing the most important goals is a key process of selection that helps focusing resources and ensuring to reach the most personally meaningful goals, and avoid goal conflict due to resource restrictions [10]. This is also consistent with empirical evidence of an age-related increase in disengagement from difficult or unattainable goals [17].

Various studies have provided empirical evidence that disengagement from unattainable goals can have positive effects on adjustment and development [18] including aspects such as personal growth [19], health [20], or subjective well-being [21]. One of the reasons for the positive effect of goal disengagement might be that disengagement alleviates goal conflict by freeing up resources [22] that can then be selectively focused on the remaining goals that, thereby, have a greater chance of being achieved. Thus, prioritizing one goal over the other in case of perceived conflict might be particularly beneficial for resolving goal conflicts, experiencing positive affect during goal pursuit, and promoting successful goal engagement.

## The present studies

The current research investigates the process of behavioral prioritization when encountering goal conflict due to limited resources. We expected that higher prioritization, as a strategy of solving goal conflict, is associated with a more positive affective experience during task pursuit as well as with lower perceived conflict between the two tasks. A plausible alternative hypothesis is that the very need to prioritize enhances the salience of the conflict between two goals [8]. However, if prioritization is an effective means to deal with goal conflict, people who prioritize should immediately profit from this prioritization through a more positive general experience of the goal pursuit. We tested these hypotheses in two studies using the same variables and the same methods. Moreover, we hypothesized that older adults prioritize more than younger and middle-aged adults.

Goal conflict was induced by giving participants two tasks but too little time to solve both. Prioritization was operationalized in two ways, namely (1) as *performance based*, indicated by the discrepancy between performance on one compared to the other task (no prioritization would be indicated by equal performance on both task), and (2) as *time based*, indicated by the discrepancy between time spend on one compared to the other task (no prioritization would be indicated by distributing the time equally among the two tasks).

Study 1 tested these hypotheses in the context of a laboratory setting in which young, middle-aged, and older adults engaged in five time-limited rounds of pursuing two goals (i.e., a word riddle and a sorting task). The limited amount of time allotted in each of the rounds was set such that it was very difficult to solve both tasks. To create a goal conflict, the instruction stressed the importance to solve both tasks. As this instruction might have inadvertently counteracted potential positive effects of prioritization by suggesting an overarching goal of solving both tasks, Study 2 used the same tasks and procedure, but changed the instruction to suggest less strongly the importance of solving both tasks. With this, we aimed at alleviating the perceived demand not to prioritize one of the goals over the other. Study 3 used self-report to test these two situations of solving goal conflict through prioritization (importance to solve both tasks more or less stressed) against each other. This experiment used hypothetical scenarios describing the instructions used in the first two experiments to test if the instructional strictness might have contributed to the hypothesized differences in results between Study 1 and 2.

## Study 1

### Methods

**Ethics statement.** The research reported in this manuscript was carried out in accordance with the regulations of the Ethics Committee of the Faculty of Arts and Sciences at the University of Zurich. These regulations define a two-stage process of ethical clearance, the first stage of which is a self-assessment of ethical risks by the researcher according to a checklist provided by the Ethics Committee. The present research passed the first stage, and therefore was exempt from further review by the Ethics Committee.

**Participants.** An *a priori* power analysis, assuming a medium age effect size of $f = .25$, an error probability of $\alpha = .05$, and a power of $1 - \beta = .90$, resulted in a required sample size of 207 participants in total. Despite our efforts to achieve the desired sample size by recruiting adults aged between 18 and 90 years out of a non-student population in the city of Zurich (Switzerland), we were only able to collect data from a sample of $N = 185$ young ($n = 67$, 18–30 years, $M = 28.47$, $SD = 3.01$), middle-aged ($n = 59$, 31–59 years, $M = 43.10$, $SD = 8.50$), and older adults ($n = 59$, 61–87 years, $M = 72.13$, $SD = 6.86$). There were more women in all age groups (younger: 59.7%, middle-aged: 69.5%, older: 55.9%). Participants did not significantly differ on self-reported health ($M = 4.63$; $SD = .91$; scale range 0–6). In line with other research, there were significant ($F[2, 180] = 8.22$, $p = .020$; $pEta^2 = .043$) age group differences in reported satisfaction with life (older: $M = 4.88$; $SD = 1.12$, middle-aged: $M = 4.36$; $SD = 1.01$, younger: $M = 4.52$; $SD = .91$; scale range 0–6).

**Procedure.** Participants were invited to the laboratory in groups of up to five, signed a consent form, filled out a short questionnaire, and were seated in front of a 20" touchscreen monitor and put on headphones for further instructions. Participants then saw a video introduction explaining the objective of each task and were given the opportunity to practice the tasks both separately and in the conflict condition until a preset criterion was met (i.e., solving each single task within 30 seconds).

Fig 1 shows a screenshot of the two tasks. In the *item sorting task* participants had to sort items and received immediate feedback on their progress by symbols indicating that the order

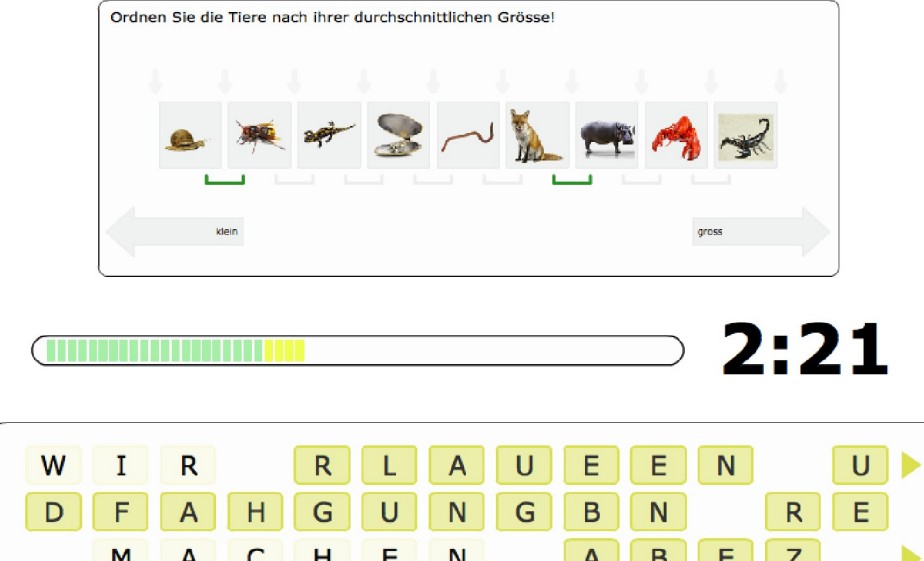

**Fig 1. Screen shot of the two tasks used to induce goal conflict.** The upper task shows the sorting task (here ranking animals according to their size) and the lower task the word riddle task (note, that the task was presented in German as participants were German-speaking adults).

of two adjacent items was correct. Based on observed performance differences in a pilot study, task difficulty was adjusted for the different age groups by using a different number of items to be sorted. In the *word riddle task* participants had to descramble a sentence by swapping single letters. Immediate feedback on task progress was given by highlighting words that were descrambled correctly. Task difficulty was adjusted for the different age groups by the extent to which the letters were scrambled as indicated by a Levenshteyn distance metric. Performance on each of the tasks was operationalized as the progress made from the original (i.e., unsorted or scrambled) condition (0%) to a perfect solution (100%).

We had selected both tasks using three criteria. First, they needed to be age-fair concerning content and evocation of interest. We hoped to achieve this by selecting one task that queries general knowledge and one that taps into verbal abilities–both skills that show little age-related decline within the studied age-range [23]. Second, we selected tasks that are unlikely to elicit interference in terms of task-switch costs. Earlier research [24] had demonstrated that switch costs become negligible if the characteristics of two competing tasks refer to a different domain of functioning (e.g., numeracy vs. word finding) compared to tasks from the same domain. Third, the tasks selected needed to be easily adaptable in difficulty in order to account for age-group differences in the speed of operating the computer. This had been established in a small pilot study with $N = 14$ younger and older adults ($n = 7$ in each age group), in which we not only measured the performance of the participants but also interviewed them about how reasonable and appealing they found the two tasks.

The core part of the study consisted of five subsequent rounds in which the two tasks were presented on the screen concurrently in randomized order and participants were asked to solve both of them within a limited period of time (240 seconds). The instruction was geared

at creating a goal conflict. Moreover, the time-based conflict between the two tasks was also made salient in that the remaining time in each round was indicated simultaneously by a count-down clock and a bar on the screen getting shorter with time running out. After the time had expired, the screen was immediately masked, so that participants were no longer able to work on the tasks.

The verbatim instruction 1 read:

"*After you have practiced each of the two tasks, it will now become more challenging. From now on, you will have to solve the two tasks simultaneously and, most notably, in limited time. There will be five rounds. Please note that the time is calculated very tightly and that you can only master the two tasks if you work fast and give this work your full attention. Even then, however, it is possible that you will not be able to finish the tasks. Nevertheless, please try to solve both tasks! It is up to you to choose the task you start with and how you proceed. Please make absolutely sure that you keep in mind the remaining time that is indicated by a bar and by a clock. When the time is over, you will not be able to continue working on the tasks.*"

After each of the rounds, participants were presented an adapted version of the Affect Mis-attribution Procedure (AMP; [25]) were asked to answer four questions on self-regulation. At the end, participants filled out another short questionnaire on constructs of self-regulation. The AMP, goal focus items, and additional self-regulation measures are not analyzed for the purpose of this article.

At the end of the session, participants were debriefed about the purpose of the study, explained that time pressure was deliberately exerted to induce goal conflict, and could choose between receiving a monetary compensation of 20 Swiss Francs in cash or to donate the respective amount to charity. The study lasted about 60 minutes.

**Measures.** *Performance*. During the five rounds, we continuously recorded on which task the participant actually worked as well as the progress they made. Averaged across the two tasks and the five rounds, younger participants showed a somewhat better performance (80.3%) compared to the middle-aged (71.7%) and the older participants (67.0%). However, on average younger and middle-aged participants also worked longer on the two tasks than older adults (younger: $M$ = 168 sec., middle-aged: $M$ = 164 sec.; older: $M$ = 132 sec.). This was possible because the three age groups differed regarding the times in which no task solving activity was recorded. This "idle time" varied from $M$ = 36 sec. in the younger group to $M$ = 44 sec. in the middle-aged group and $M$ = 62 sec. in the older group. When statistically control-ling for "idle time," the difference in performance between the younger (74.3%), the middle-aged (70.0%), and the older group (75.6%) was not significant ($F$[2, 181] = 2.01, $p$ = .14).

*Prioritization*. We computed two indicators of task prioritization. The first one represents *performance-based prioritization* as the discrepancy in performance between the two tasks, with higher values indicating prioritization, in that one task was solved better than the other. For each round, we computed the absolute value of the difference between the $z$-standardized performance score on one task and the z-standardized performance score on the other task. Using these five indicators, we then modeled performance-based prioritization as a latent con-struct and obtained a measurement model that fit the data without any significant deviation ($\chi^2$[13] = 10.63, $p$ = .64; RMSEA = .000; TLI = 1.00). An exploratory extension of the measure-ment model to a latent growth model with a linear slope component did not significantly improve the model fit.

The second indicator represents *time-based prioritization* as the discrepancy of time spent on each of the tasks, with higher values indicating that participants worked longer on one task

than on the other. For this indicator we computed the absolute value of the difference between the z-standardized time worked on one task and the z-standardized time worked on the other task for each of the five rounds. We then modeled time-based prioritization as a latent construct in the same way as we did for performance-based prioritization. The measurement model fitted the data without any significant deviation ($\chi^2[13] = 9.82$, $p = .71$; RMSEA = .000; TLI = 1.00). An exploratory extension of the measurement model to a latent growth model with a linear slope component did not significantly improve the model fit. Performance-based prioritization was only weakly correlated with time-based prioritization ($r = .18$).

*Affective experience*. After one third of the time had expired in each round, we assessed the affective experience of the participants with the Self-Assessment Manikin (SAM) [26]. Participants were asked to rate their current mood on the affective dimensions of valence (anchors were labeled "bad" and"good"), arousal (anchors were labeled "calm" and"excited"), and control (anchors were labeled "nothing under control" and "everything under control"). We modeled all three dimensions in one single model as separate latent growth processes. The best fitting model had an intercept and a slope component for valence and arousal, and an intercept only for control ($\chi^2[85] = 132.78$, $p < .001$; RMSEA = .056; TLI = .96). An inspection of overall means suggested that valence did not change over time ($\alpha = .04$, $CI_{95} = [.00;.08]$, $p = .27$) and that arousal significantly decreased ($\alpha = -.11$, CI95 = [-0.19; -0.03], $p = .013$). Older participants had a less pronounced slope of valence ($\beta = -.30$, CI95 = [-0.59; -0.01], $p = .043$) and a lower intercept of control ($\beta = -.35$, CI95 = [-0.51; -0.19], $p < .001$). Otherwise, the three age groups did not differ on any of the other growth parameters.

*Perceived task conflict*. After two thirds of the time has expired in each round, participants were asked to rate "How much does working on one task conflict with working on the other?" on an 11-point-scale with the anchors of 0% and 100%. We modeled perceived conflict as a linear latent growth process, which fitted the data without significant deviation ($\chi^2[10] = 11.63$, $p = .31$; RMSEA = .030; TLI = 1.00). The slope component had a significant mean ($\alpha = -.01$, CI95 = [-0.02; 0], $p = .025$) and significant variance ($p < .001$). Middle-aged ($\beta = .27$, CI95 = [0.09; 0.45], $p = .002$) and older adults ($\beta = .25$, CI95 = [0.07; 0.43], $p = .005$) had higher intercepts of perceived conflict than younger adults as the reference group. However, there were no significant age-group differences in the slope of perceived conflict ($.06 < p < .74$). This means that age-groups experienced the same amount of increase in their perceived conflict over the course of the experiment.

**Data-analytic approach.** In order to account for the multiple measurements across the five rounds of the experiment, we modeled both the exogeneous (i.e., prioritization) and the endogenous variables (i.e., perceived task conflict and affective experience) as latent growth models including an intercept and, whenever possible, also a slope of the five subsequent measurements. Age-group effects were determined by regressing the intercept and/or slope of the variable under investigation on the grouping variable. We refrained from computing multi-group models (that would have allowed testing additional hypotheses) due to the moderate size of our samples.

## Results

**Age-group differences.** In order to test for age-group differences, we regressed the two latent indicators of task prioritization on the two age-group dummy variables with the youngest group as the reference group ($\chi^2[61] = 74.33$, $p = .12$; RMSEA = .034; TLI = .92). Results with regard to performance-based prioritization show that the younger and the middle-aged adults did not differ significantly ($\beta = .14$, CI95 = [-0.06; 0.34], $p = .15$) but that the younger and the older adults did ($\beta = .45$, CI95 = [0.27; 0.63], $p < .001$; see upper panel Fig 2).

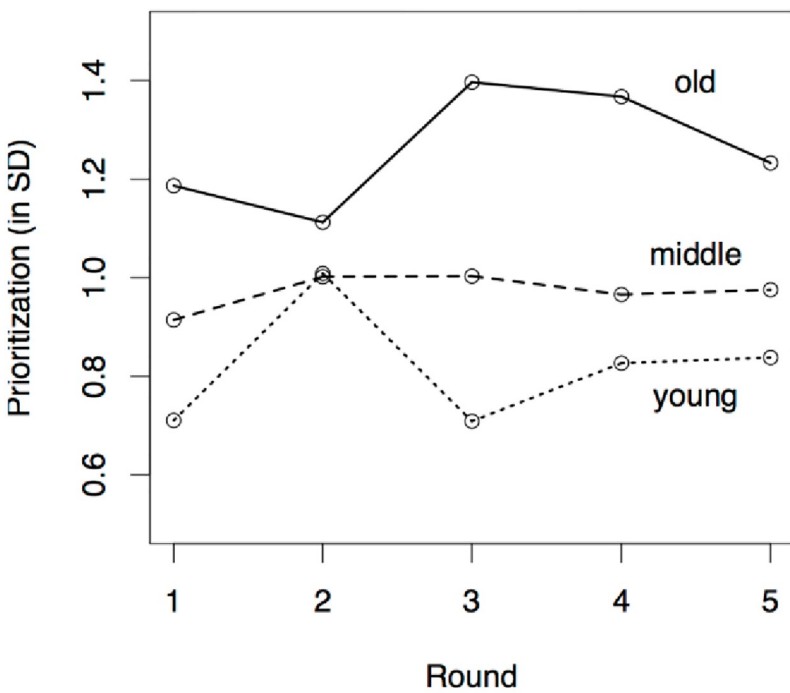

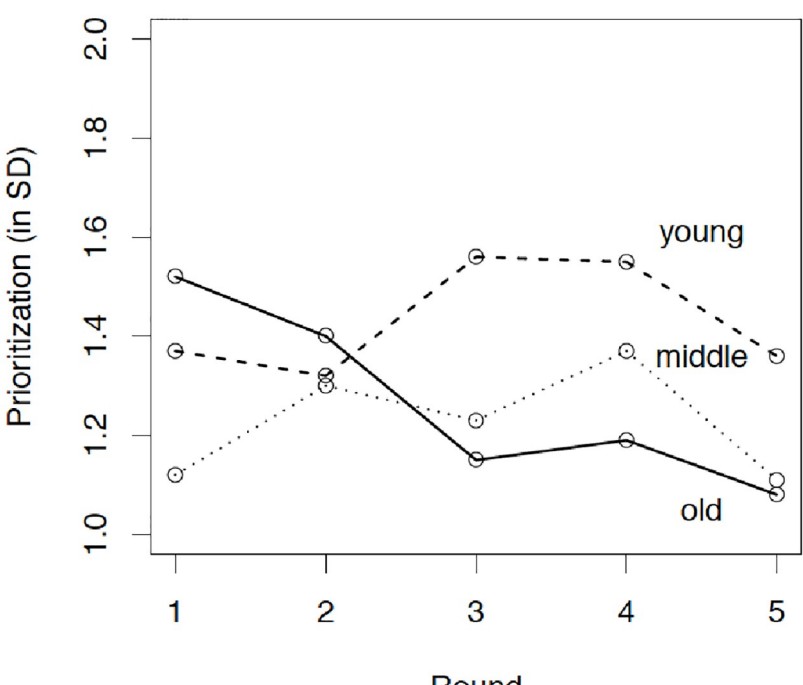

**Fig 2. Age-related differences in performance-based prioritization (upper panel) and time-based prioritization (lower panel) over the course of the five rounds in Study 1.**

Regarding time-based prioritization (see lower panel in Fig 2), the middle-aged group was marginally higher than the younger group (β = .22, CI95 = [-0.02; 0.46], $p$ = .06) but the older group did not differ from the younger (β = .03, CI95 = [-0.21; 0.27], $p$ = .84). Thus, our hypothesis of an age-related increase in prioritization was only supported regarding performance-based prioritization.

**Associations with affective experience.** To keep model complexity manageable, we computed separate models for the two latent indicators of task prioritization and each of the three latent indicators of affective experience. The model for valence fitted the data well ($\chi^2$[101] = 109.94, $p$ = .26; RMSEA = .022; TLI = .99). Against our expectations, the intercept of valence was negatively associated with performance-based prioritization ($r$ = -.39, CI95 = [-0.59; -0.19], $p < .001$); valence was not significantly associated with time-based prioritization but the correlation coefficient pointed in the same direction ($r$ = -.20, CI95 = [-0.04; 0.44], $p$ = .11). Furthermore, the slope of valence was marginally significantly correlated with performance-based prioritization ($r$ = -.39, CI95 = [-0.78; 0], $p$ = .06). In other words, participants who prioritized more in terms of performance reported a more negative affective experience, which was likely to deteriorate over the course of the five rounds.

The model for arousal fitted the data very well ($\chi^2$[101] = 101.31, $p$ = .42; RMSEA = .004; TLI = 1.00). However, neither the intercept nor the slope component of arousal was significantly associated with any kind of task prioritization ($.26 < p < .87$). Hence, our hypothesis that prioritization leads to less arousal (by resolving conflict) has to be rejected.

The model for control showed a satisfactory fit ($\chi^2$[106] = 132.38, $p$ = .04; RMSEA = .037; TLI = .95). The intercept of control was significantly negatively correlated with performance-based prioritization ($r$ = -.34, CI95 = [-0.50; -0.18], $p < .001$); the negative association of control and time-based prioritization pointed in the same direction but was not statistically significant ($r$ = -.13, CI95 = [-0.29; 0.03], $p$ = .10). As in the case of valence, the associations had the opposite direction as expected with higher prioritization being associated with feeling less control.

**Associations with perceived task conflict.** A joint model with the two latent indicators of task prioritization and the latent indicator of perceived task conflict was set up. The model fitted the data well ($\chi^2$[101] = 115.08, $p$ = .16; RMSEA = .027; TLI = 0.98). The intercept of perceived task conflict was not correlated significantly with performance-based prioritization ($r$ = .16, CI95 = [-0.04; 0.36], $p$ = .13) and time-based prioritization ($r$ = .12, CI95 = [-0.12; 0.36] $p$ = .32). However, the slope of perceived task conflict was significantly correlated with performance-based prioritization ($r$ = .38, CI95 = [0.14; 0.62], $p$ = .001) and marginally significantly with time-based prioritization ($r$ = .23, CI95 = [-0.04; 0.50], $p$ = .09). Contrary to our hypotheses, the significant associations were positive. Thus, higher prioritization was associated with an increase in perceived conflict over the course of the five rounds (see Fig 3).

## Discussion

Study 1 investigated age-related differences in solving goal conflict due to limited resources by prioritizing one goal over another. Based on previous research on age-related differences in conflict between higher-order personal goals (such as wanting to start regular physical exercise; [14]), we had expected that older adults are more likely than younger or middle-aged adults to prioritize one goal over another when confronted with two conflicting goals. The current study confirmed this expectation for performance-based prioritization. Older adults showed higher performance-based prioritization compared to younger adults. However, they did not differ from middle-aged adults. It seems that, when confronted with two conflicting goals, older adults selectively optimize their performance more so than younger adults do.

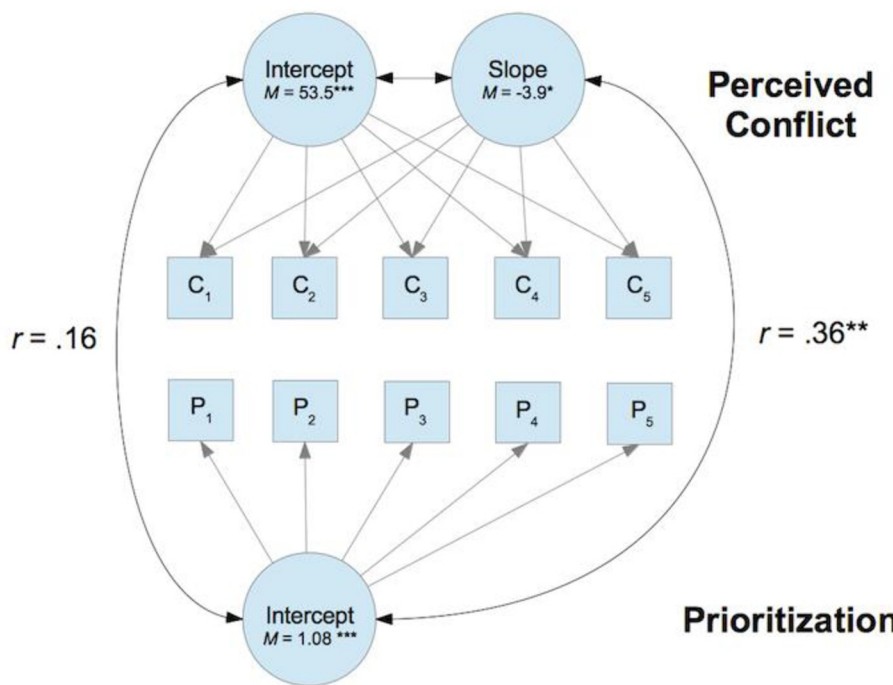

**Fig 3. Simplified model depiction of the relation of perceived conflict (C) and prioritization (P) over the course of the five rounds Study 1.** Contrary to expectations, prioritization was positively related to perceived conflict.

This finding fits with predictions of the model of selection, optimization, and compensation [15] that posits that due to declining resources, older adults are more selective regarding the goals they attempt to optimize.

Our central prediction was that prioritization is associated with experiencing lower conflict and more positive affect, less arousal, and more control. However, the opposite pattern emerged: The more participants prioritized one goal over the other, the more conflict they experienced over the course of goal pursuit, the less positive their affective experience, and the less they felt in control.

This is a perplexing result that contrasts with the previous research showing that self-reported disengagement from blocked personal goals and re-engagement in other personal goals is positively related to various indicators of adaptive functioning [18, 20, 21]. However, these studies investigated disengagement on a self-reported level and with regard to higher-order personal goals. In contrast, the current study operationalized prioritization using a behavioral indicator. Moreover, the current study used the lower-level goals of having to solve five sets of knowledge-related tasks over the course of a total of 20 minutes.

We had instructed participants to solve *both* tasks (in order to increase the perceived conflict between the goals) and thereby probably created a superordinate goal. Hence, when prioritizing one goal over the other despite having been instructed to solve both tasks, participants might have felt that the very act of prioritizing lead to higher conflict with the relatively neglected goal, which might have also led to more negative affect and to feeling less in control. This finding is in line with results by Ballard et al. [9] who found that participants attempted to achieve two goals competing for time and effort, even if this did not guarantee that they maximized their financial rewards. Thus, people might be generally inclined to attempt to achieve all of the goals in a given situation, and stressing the importance to do so as we did in Study 1, might have even strengthened this effect.

## Study 2

Study 2 was conducted to investigate if results replicate when deemphasizing that both tasks ought to be solved, thereby easing the strong demand we might have created in Study 1 not to prioritize one task over the other. Given the difficulties we had in Study 1 to recruit a sufficient number of middle-aged adults and the lack of systematic differences between middle-aged and the older adults on prioritization, Study 2 included only younger and older adults. Other than that, Study 2 used the same measures and tasks as Study 1 with two exceptions concerning the instruction and the presentation of the tasks that are reported below.

### Methods

**Participants.** Younger and older adults were recruited from a participant pool of the Life-Management Laboratory at the University of Zurich (Switzerland). Using the same logic for the power analysis as in Study 1, we aimed at recruiting $n = 69$ for each age group of younger and older adults from a non-student population. Unfortunately, however, it seemed that we had exhausted the number of younger and older non-student volunteers, so that we did not quite reach this number despite our efforts. After excluding four participants who did not understand the tasks or did not comply with the instructions, the sample comprised a total of $N = 117$ younger and older adults (younger adults: $n = 61$, 18–30 years, $M = 23.95$, $SD = 3.13$; $n = 56$ older adults, 61–87 years, $M = 71.18$, $SD = 6.27$). Females were overrepresented both in the younger (70.5%) and in the older group (60.7%).

Participants in the two age groups did not differ significantly regarding their self-reported health, which was overall good ($M = 4.70$; $SD = .88$; scale range 0–6). However, younger participants reported a significantly ($F[1, 115] = 9.36$, $p = .003$; $pEta^2 = .072$) poorer satisfaction with life ($M = 4.54$; $SD = .92$) compared to the older participants ($M = 5.11$; $SD = 1.11$; scale range 0–6).

**Procedure and measures.** The procedure, tasks, and measures were identical to Study 1. However, different to Study 1, the instruction now had no reference to the importance of solving both tasks simultaneously. Moreover, the colors and contrast of the task on which participants were not working at the time were muted such that they seemed to disappear more into background. Only when the mouse was used to activate the "dormant" task did it again become clearly visible. This was done in order to deemphasize the salience of the non-focal task at any point in time. Participation in the study took about 60 minutes.

*Performance.* Averaged across the two tasks and the five rounds, younger participants showed a somewhat better performance (78.7%) compared to the older participants (66.7%). As in Study 1, younger participants also worked on average longer on the two tasks ($M = 188$ sec.) than older participants ($M = 148$ sec.). This is partly due to the fact that older participants needed longer for the interspersed assessments of perceived conflict and affect ($M = 18$ seconds) than younger participants ($M = 8$ seconds). When statistically controlling for the time to respond to these items, there was no significant difference in performance between the younger (72.9%) and the older group (73.0%; $F[1, 114] = .78$, $p = .38$).

*Prioritization.* We computed the same two indicators of prioritization as in Study 1 based on the performance measures and the time needed to solve the two tasks. For both *performance-based prioritization* ($\chi^2[13] = 17.00$, $p = .20$; RMSEA = .051; TLI = .89) and *time-based prioritization* ($\chi^2[13] = 16.43$, $p = .23$; RMSEA = .047; TLI = .94) the measurement models fitted the data without any significant deviation. Different to Study 1, time-based prioritization now substantially correlated with performance-based prioritization ($r = .69$), presumably because the time measure was now more precise due to the visual muting of the non-active task.

*Affective experience*. We again assessed participants' current affect (*valence*, *arousal*, and *control*) using the SAM and modeled all three dimensions in one single model as separate linear latent growth processes, which described the data very well ($\chi^2$[78] = 91.67, *p* = .14; RMSEA = .040; TLI = .98). An inspection of overall means suggested that valence did not change over time ($\alpha$ = .01, CI95 = [-0.07; 0.09], *p* = .90), that arousal significantly decreased ($\alpha$ = -.19, CI95 = [-0.31; -0.07], *p* = .001), and control significantly increased ($\alpha$ = .13, CI95 = [0.13; 0.13], *p* = .003). There was also some evidence for variance in the slope components of valence (*p* = .08), arousal (*p* < .001), and dominance (*p* = .10).

As in Study 1, older adults had a less pronounced slope of valence ($\beta$ = -.37, CI95 = [-0.74; -.01], *p* = .049) and a lower intercept of control ($\beta$ = -.27, CI95 = [-0.47; -0.07], *p* = .008). Otherwise, there were no other significant age group differences on this measure.

*Perceived task conflict*. Conflict was assessed using the same single item as in Study 1. We modeled perceived conflict as a linear latent growth process, which fitted the data satisfactorily ($\chi^2$[10] = 17.64, *p* = .06; RMSEA = .083; TLI = .98). Although the slope component did not have a significant mean for the overall sample ($\alpha$ = -.01, CI95 = [-0.03; 0.01], *p* = .12), it had sufficient variance (*p* = .008). Younger and older participants did not significantly differ on any of the growth parameters for perceived task conflict (.36 < *p* < .50).

## Results

Data were analyzed in the same way as in Study 1.

**Age group differences.**   In order to test the hypothesized age-group differences, we regressed the two latent indicators of task prioritization on age group using a joint model, which fitted the data very well ($\chi^2$[53] = 52.42, *p* = .50; RMSEA = .000; TLI = 1.00). Age group significantly predicted performance-based prioritization ($\beta$ = .41, CI95 = [0.17; 0.65], *p* = .001) but not time-based prioritization ($\beta$ = .17, CI95 = [-0.07; 0.41], *p* = .16; see Fig 4). Thus, replicating results of Study 1 and despite the relatively high correlation between the two operationalizations, the hypothesis that older adults prioritize more than younger adults when faced with conflicting goals was again supported only for performance-based prioritization.

**Associations with affective experience.**   The model for valence showed a satisfactory fit ($\chi^2$[101] = 135.40, *p* = .01; RMSEA = .054; TLI = .92). The intercept of valence was marginally significantly negatively associated with both performance-based prioritization (*r* = -.27, CI95 = [-0.56; 0.02], *p* = .06) and time-based prioritization (*r* = -.27, CI95 = [-0.54; .00], *p* = .06). The slope of valence was not significantly correlated with any of the indicators of prioritization. When fixing these three correlations to zero, the model fit did not significantly deteriorate ($\Delta\chi^2$[2] = 1.02, *p* = .60; $\Delta$RMSEA = .001; $\Delta$TLI = .00). In this reduced model, the intercept of valence was significantly negatively associated with both performance-based prioritization (*r* = -.21, CI95 = [-0.41; -0.01], *p* = .04) and time-based prioritization (*r* = -.28, CI95 = [-0.52; -0.04], *p* = .02). This replicates the results of Study 1 that prioritization is associated with less positive affect.

The model for arousal showed a very good fit to the data ($\chi^2$[101] = 100.94, *p* = .48; RMSEA = .000; TLI = 1.00). Neither the intercept nor the slope component of arousal was significantly associated with task prioritization. Fixing the slope correlations to zero, as we did for valence, did not change this overall pattern. Again replicating results of Study 1, arousal appears to be unrelated to prioritization.

The model for control showed a just satisfactory fit ($\chi^2$[101] = 144.44, *p* = .003; RMSEA = .061; TLI = .89). The intercept of control was significantly correlated with time-based prioritization (*r* = -.28, CI95 = [-0.55; -0.01], *p* = .05) but not with performance-based prioritization (*r* = -.16, CI95 = [-0.45; 0.13], *p* = .29). No significant correlations emerged with the slope

## Study 2

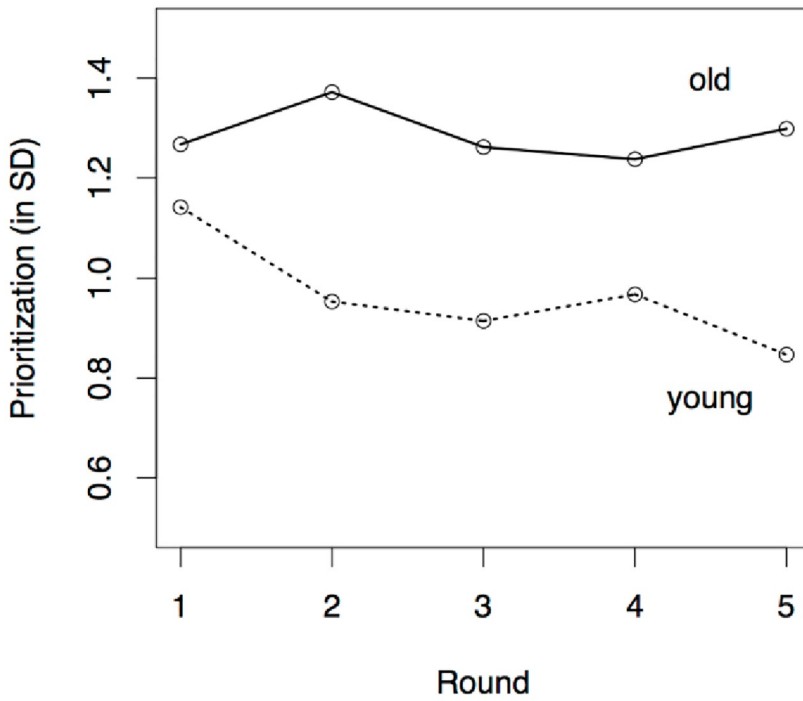

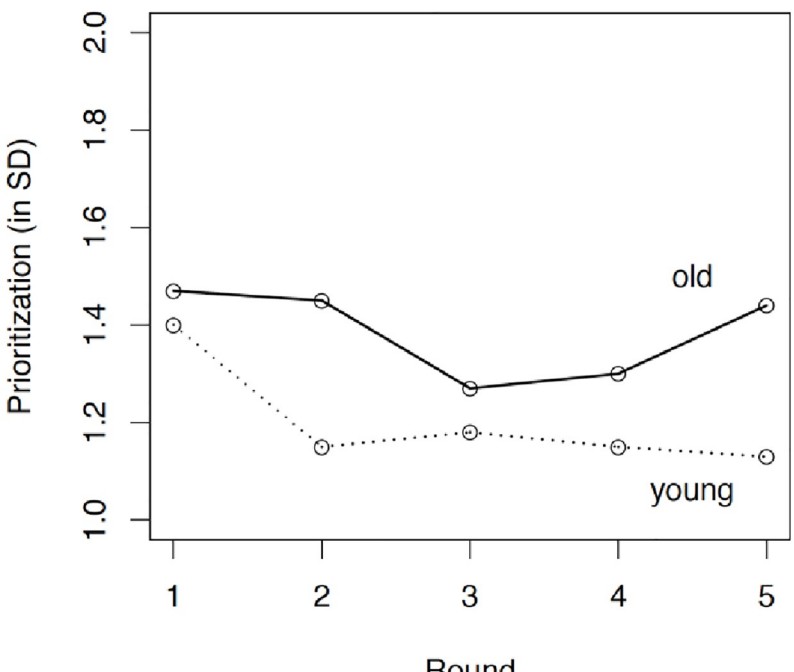

**Fig 4. Age-related differences in performance-based prioritization (upper panel) and time-based prioritization (lower panel) over the course of the five rounds in Study 2.**

component. Fixing these correlations to zero did not deteriorate model fit in significantly ($\Delta\chi^2[2] = .29$, $p = .87$; $\Delta$RMSEA = -.002; $\Delta$TLI = -.01). In this reduced model, the intercept of control was negatively associated with time-based prioritization ($r = -.30$, CI95 = [-0.55; -0.05], $p = .02$) but not with performance-based prioritization ($r = -.19$, CI95 = [-0.46; 0.08], $p = .17$). Again, this replicates the results of Study 1 regarding the association between higher prioritization and lower control.

**Associations with perceived task conflict.** A joint model with the two latent indicators of task prioritization and the latent indicator of perceived task conflict was set up and fitted the data very well ($\chi^2[101] = 103.48$, $p = .41$; RMSEA = .014; TLI = 0.99). The intercept of perceived task conflict was unrelated to performance-based prioritization ($r = .11$, CI95 = [-0.20; 0.42], $p = .49$) but significantly positively correlated with time-based prioritization ($r = .34$, CI95 = [0.07; 0.61], $p = .01$). Above that, there were no significant associations between the indicators of prioritization and the slope of perceived conflict. Hence, against our hypotheses but replicating Study 1, time-based prioritizing seems to be associated with higher and not lower perceived task conflict.

**Associations with affective task preference.** The model with two latent indicators of task prioritization and one latent indicator for affective task preference fit the data very well ($\chi^2[98] = 95.77$, $p = .54$; RMSEA = .000; TLI = 1.00). Affective task preference was not significantly correlated with performance-based prioritization ($r = .19$, CI95 = [-0.10; 0.48], $p = .21$) but was so with time-based prioritization ($r = .29$, CI95 = [0.04; 0.54], $p = .03$). Thus, as expected, spending more time on one task than the other is associated with a stronger affective preference for one over the other task on the implicit level.

## Discussion

Study 2 replicated some key findings from Study 1, but also yielded some important differences. First, we found again that performance-based prioritization of one task over the other that was stronger for older compared to younger participants. Using an instruction different to Study 1 that did no longer stress the importance of solving both tasks, we found that higher performance-based prioritization was no longer negatively associated with affective experience or perceived conflict. Time-based prioritization, however, was again associated with experiencing higher conflict. Taken together, we interpret these findings as indicating that prioritization shows negative effects on affect and perceived conflict when people are asked to perform equally well on two tasks and are hence likely to form a superordinate goal of "solve both tasks" [9], but show differences in task performance. As this pressure is lessened, performance discrepancies between the two tasks are no longer associated with perceived task conflict.

Note, however, that prioritizing one task over the other in terms of the time spent on each of the two tasks was related to experiencing more conflict even under a more lenient instruction. The high salience of time for solving the tasks (the passage of and the remaining time was continuously and prominently displayed on the screen) might have implied to people that they ought to give both tasks an equal amount of time, such that that focusing one's time more on one of them might have caused them to perceive higher conflict. In other words, we eased the instruction of having to solve both tasks equally well in but did not change the salience of time constraints in Study 2.

## Study 3

Because participants were not randomly assigned to the different conditions of Study 1 and 2, our interpretation of the different findings between the two Studies remains speculative. Therefore, we conducted a self-report follow-up study where we presented participants with

hypothetical scenarios describing the experimental setting of Studies 1 and 2, and asked them to imagine the respective situations, and then either that they prioritized or divided their effort between the two tasks. To mimic the two studies as closely as possible, we used a between-subjects approach (so as to not to create a direct comparison in the experimental set-up for the participants) with the following factors: 2 (study condition: stressing the importance of achieving both goals vs. not) x 2 (prioritization: focusing on one goal vs. dividing effort between two goals). We hypothesized that participants in the condition that stressed the importance of achieving both goals expect to feel more conflicted and experience more negative affect when prioritizing one goal over the other than when dividing their effort between the two goals.

## Methods

**Participants.**    Given the large number of participants necessary for this follow-up study and the fact that we had already exhausted the potential participant pool in Zurich and its vicinity, we ran this study as an online study using Amazon's Mechanical Turk (MTurk). Of the recruited participants, $n = 166$ quit the questionnaire after a few questions or missed one or more points of the inclusion criteria (age < 18 years; USA resident; correct responses to check questions about their age). The final sample included $N = 721$ adults, 48% females and about equal numbers in each age group of young ($n = 243$, 20–30 years, $M = 26.6$, $SD = 2.63$), middle-aged ($n = 240$, 31–60 years, $M = 41$, $SD = 8.07$), and older adults ($n = 238$, 61–86 years, $M = 68.4$, $SD = 4.9$). The age groups did not differ regarding subjective health, $F(2, 718) = 1.1$, $p = .33$, but life satisfaction ("Overall, how satisfied are you with your life?", scale: 0–6) was higher in the older adults ($M = 4.42$, $SD = 1.4$) than in the younger ($M = 3.94$, $SD = 1.48$) or middle-aged adults, ($M = 3.94$ $SD = 1.28$), $F(2, 718) = 8.88$, $p < .001$. This is consistent with the other two studies.

**Procedure and measures.**    Participants read one of four vignettes describing the procedure of Study 1 or Study 2. We asked participants to imagine the situation as well as their choice to either focus on one of the two tasks or divide their efforts between them. This resulted in four conditions that were randomly assigned to participants, varying the degree to which it was stressed that both tasks were equally important and the prioritization in pursuing the two tasks. The vignette mimicking Study 1 with high (vs. low prioritization in parenthesis) read:

> *Imagine the following: You volunteer for a research study taking place in the Psychology Department at a University. You will be reimbursed for participation. For the next half hour, you have to work on two goals. The time seems too short to accomplish both of them. The instruction stresses that both goals are equally important. [You decide to focus on one of the goals at the cost of the other OR You decide to divide your time and energy between the two goals.] Both goals are displayed on the computer screen during the entire time. You see the clock on the computer screen ticking away.*

The vignettes mimicking Study 2 was identical with the exception that the sentence "*The instruction stresses that both goals are equally important*" was replaced with "*The instruction does not specify the relative importance of the goals.*"

Participants indicated how they would feel in this situation on the eight adjectives conflicted, engaged, tense, interested, calm, nervous, relaxed, stressed (response scale ranging from 0 to 5) and the same SAM as in Study 1 and 2 assessing valence, arousal, and control. Based on their intercorrelations, we aggregated "stressed," "nervous," and "tense" ($\alpha = .91$, $M = 3.9$; denoted as "stressed"), as well as "calm" and "relaxed" ($\alpha = .89$, $M = 3.1$; denoted as "calm").

## Results

A 2 x 2 x 3 between-subjects MANOVA tested whether the imagined response to either prioritize one goal over the other vs. to divide one's efforts between the two goals differed by the situation describing Study 1 vs. Study 2 and by age group regarding the imagined affective reactions of feeling conflicted, stressed, calm, engaged, and interested as well as regarding the SAM dimensions valence, arousal, and control. There were significant main effects of Study scenario ($F[8, 702] = 6.03$, $p < .001$; $pEta^2 = .065$), prioritization ($F[8, 702] = 7.13$, $p < .001$; $pEta^2 = .075$), age group ($F[8, 702] = 3.59$, $p < .001$; $pEta^2 = .039$), and interactions of described Study scenario and prioritization ($F[8, 702] = 477$, $p < .001$; $pEta^2 = .052$). No other interactions were significant (all $F[16, 1406] < 0.6$, $p \geq .57$). As expected, for the vignettes describing Study 1 stressing the importance of solving both tasks, participants imagined to feel more stressed, less calm, more engaged, and more aroused than for the vignettes describing Study 2 (all $F(1, 709) > 8.5$, all $p \leq .003$). However, there were no differences in how conflicted or interested they expected to feel, or in valence or control ratings (all $F(1, 709) \leq 0.08$, all $p \geq .14$).

There were only two significant effects of the imagined prioritization of one goal over the other (vs. dividing one's efforts between the two goals), namely on engagement ($F[1, 709] = 47.2$, $p < .001$; $pEta^2 = .062$) and interest ($F[1, 709] = 7.39$, $p = .007$; $pEta^2 = .01$). For all other outcomes, including how conflicted participants expected to feel, there were not significant differences between the imagined prioritization or division of effort, all $F(1, 709) \leq 0.08$, all $p \geq .33$.

Both of the significant main effects or prioritization were qualified by an interaction with the described study scenario; for engagement $F[1, 709] = 17.2$, $p = .001$; $pEta^2 = .014$) and interest ($F[1, 709] = 10.68$, $p = .001$; $pEta^2 = .015$). Engagement was expected to be highest in the situation where both goals were stressed to be equally important and people divided their efforts between the two tasks ($M = 4.91$, $SD = 1.14$) and lowest when imagining to participate in Study 2 and prioritizing one task over the other ($M = 3.73$, $SD = 1.65$). The expected engagement for the other two conditions were between those extremes (Study 1, prioritization: $M = 4.57$, $SD = 1.17$; Study 2, division of effort: $M = 4.69$, $SD = 1.14$). For interest, the difference occurred in the imagined Study 1, where interest was expected to be highest when dividing efforts between the two goals ($M = 4.89$, $SD = 1.11$) and lowest when prioritizing ($M = 4.38$, $SD = 1.27$); expected interest in Study 2 did not differ regarding prioritization ($M = 4.63$, $SD = 1.14$ and $M = 4.68$, $SD = 1.11$).

There were significant age-group differences regarding expected conflict (decreasing with age), stress (decreasing with age), calmness (increasing with age), and regarding valence (more positive with age) and control (higher with age), $F(2, 709) > 5.9$, all $p \leq .003$. There were no age differences regarding expected engagement or arousal, all $F(2, 709) \leq 2.5$, all $p \geq .07$.

Taken together, the follow-up self-report study supported our interpretation of the differences between the behavioral Studies 1 and 2. Participants imagined feeling more stressed, less calm, more engaged, and more aroused in a situation describing Study 1, namely when it was stressed that two tasks are equally important but there is not sufficient time to accomplish both tasks. Interestingly, there were no differences in how conflicted participants expected to feel in these two scenarios.

Different to the results of the behavioral studies, the self-report study did not show that people imagine that they might feel more conflicted when they prioritize one task over the other. To the contrary, they seemed to infer that they would be less engaged and interested when prioritizing compared to attempting to solve both tasks.

## General discussion

Prioritization has been hailed as one possibility to manage goal conflict [8, 14]. Contrary to our hypothesis, however, performance-based prioritization did not help to ease the experience of goal conflict. Instead, when the importance of achieving both of the competing goals was stressed, performance-based prioritization was associated with even stronger perceived goal conflict over the course of goal pursuit. Similarly, prioritization was associated with more negative affect ratings and lower perceived control, and did not decrease arousal. As a short-term strategy to alleviate the negative psychological effects of pursuing two conflicting, equally important goals, prioritization does not appear to be particularly effective [8]. This finding is consistent with the position by Ballard et al. [9] who posit that the achievement of a given goal carries in itself a high value for people. Thereby, prioritization might have negative emotional consequences that counteract the potential positive consequences due to the freeing of resources. Taken together, in this setting and in the short term, the psychological "costs" of prioritization seem to outweigh its "benefits." Study 3 supports this interpretation.

In the current short-term study design, perceived conflict might have served as a trigger of prioritization and not as its psychological consequence. People (of all ages) need to perceive a situation as conflicting first in order to respond to this situation with strategies such as behavioral prioritization. We cannot rule out this interpretation but the results of Study 2, in which more conflict was *not* a consequence of prioritization, indirectly support this interpretation. Prioritization during the pursuit of two simultaneously presented tasks might not have reduced the perceived conflict because this specific set-up of both tasks on one computer screen reminds people throughout working on one of the tasks that they are doing so at the cost of the other, neglected task. This might have prevented goal shielding, a mechanism enabling temporary prioritization of the currently pursued goal [7].

Another reason why we might not have found the hypothesized negative association between prioritization and perceived conflict lies in the particular set-up of creating goal conflict used in Study 1. A time-based conflict of having to solve two tasks within a very short period of time might be similar to the situation in which younger and middle-aged adults often find themselves in the educational or work context. During young and middle adulthood in particular, conflicting goals cannot easily be given up even if prioritization would solve the conflict. For instance, founding a family and establishing a professional career are two goal domains that often conflict in young adulthood, just as taking care of one's family and maintaining one's job performance often draws on the same resource of time in middle-adulthood [27]. These conflicts typically negatively affect indicators of well-being and health [6]. However, giving one of these domains up or clearly prioritizing one over the other might not be an option as both seem vital to most young and middle-aged adults. In fact, prioritization of work over family might intensify feelings of guilt and conflict regarding one's family obligations, and vice versa. The current study might have created a similar situation by instructing participants to pursue both tasks, and hence made it more difficult for younger and middle-aged than for older adults to prioritize one task over the other.

In order to create time-based conflict, the instruction of Study 1 stressed that both tasks had to be solved within a limited time frame. All age groups including the older participants might have understood pursuing both tasks as a superordinate goal they had to follow in order to comply with the study instructions. Thus, prioritization could not fully resolve the conflict as it runs counter the superordinate goal of solving both tasks. This might have led to the negative association of prioritization with perceived conflict and affective experiences. Dampening the instruction regarding the importance of performing equally well in Study 2 indeed eliminated this effect. Interestingly, given that we did not change the salience of the limitation and

passing of time during goal pursuit, we did not seem to ease the effect of time-based prioritization on perceived conflict. The follow-up self-report Study 3 asking participants to imagine being in situation either mimicking Study 1 or Study 2, showed that people do in fact expect to feel more stressed, less calm, more engaged, and more aroused in a situation that stresses the importance of solving two tasks without having sufficient time to do so. This result supports our interpretation of the differences between Studies 1 and 2. Note, that the samples of the three studies were not entirely equivalent in terms of some of the background variables. Such sample variations in measures of imperfect reliability are to be expected.

Converging with prior studies, we found that performance-based prioritization increases with age. To our knowledge, this is the first study to investigate prioritization behaviorally in a controlled laboratory setting. Extant studies have asked participants to report how easily they generally disengage from blocked goals [22] or had participants rate the degree to which their personal goals serve the same superordinate goals [14]. On the one hand, this assessment raises the question of how well self-report reflects what people actually do when they are faced with blocked or conflicting goals. On the other hand, it leaves open if disengagement or prioritization are the cause of resolving conflict or rather their result. A person who can no longer pursue a given goal because it is blocked or conflicts with other goals might then reconstruct having had to give up the goal as disengagement or prioritization.

The current studies attempted to get closer to the *process* of prioritization when faced with goal conflict by assessing behavioral prioritization over the course of pursuing two conflicting goals over time. Moreover, the conflict was induced by limiting an essential resource for pursuing the goals in order to mimic one of the main sources of goal conflict [10]. Both goals were given to the participants, thereby ruling out that people might have already differed in selecting more congruent goals. Both goals were carefully calibrated in that there were virtually no performance differences between the age groups. This is important, as the findings regarding age-related differences in prioritization cannot be attributed to differences in task performance. However, the control over inducing conflict and being able to measure prioritization behaviorally and over time also necessitated the use artificial tasks that were probably not very personally important to the participants. This limits the generalizability of the results to the pursuit of personal goals in everyday life.

One of the general challenges in research on personal goals is that they are either personally relevant or experimentally controlled, but not both: When goals are induced in a well-controlled laboratory setting, the goals are short-term and typically not personally important. In contrast, when investigating goals in real life [2, 14], goals are personally important but differ in such dimensions as difficulty and concreteness, and performance measures cannot be easily compared across goals. The current set of studies aimed at complementing the extant research on prioritization and goal disengagement based on self-report measures concerning people's personal goals with a well-controlled laboratory task designed to assess the process of prioritization behaviorally.

## Conclusion

To our knowledge, the current studies are the first to investigate the process of prioritization behaviorally and during the time of the goal conflict. Results suggest that older adults show more performance-based prioritization than younger adults when faced with conflicting goals. This is in line with theoretical notions expecting a higher behavioral selectivity in older adults [28] and empirical findings demonstrating this selectivity in ecologically valid studies [13]. The effect replicated across two independent samples. However, more research is needed to investigate the association between prioritization on the one hand, and psychological

outcomes such as perceived conflict or affective experiences on the other. Contrary to our hypotheses, theoretical propositions in the literature [22], and empirical research [18] suggesting the benefits of disengagement vis-à-vis unattainable goals or in the case of goal conflict [14], we did not find that prioritization pays off while pursuing the prioritized goal.

Note, however, that the studies that have found a benefit of prioritization and disengagement are based on goals encompassing considerably longer time frames. Importantly, as of yet these studies have not focused on the *process* of disengagement itself. Finally, prior studies on disengagement have investigated coping with blocked or unattainable goals, whereas our studies used the less dramatic sample case of goal conflict. Disengagement might be the most adaptive strategy when goals are permanently blocked, but lead to a stronger experience of goal conflict when people still harbor the hope to eventually achieve both goals. Based on the current set of studies, future research on prioritization needs to carefully consider (a) the source of the need to prioritize (i.e., blocked goals, goal conflict) as well as (b) the time frame when investigating the dynamics of goal disengagement.

## Context

This article builds on the prior research by the first author that shows that older adults report fewer goal conflicts despite the decrease in goal-relevant means compared to younger adults [2, 11, 14]) and the literature showing that older adults report in questionnaire studies that they disengage from blocked or unattainable goals more easily [20, 22]. Integrating these two lines of research in light of the motivational literature that assumes that temporarily disengaging from one goal in order to prioritize another goal might be one of the main processes to alleviate goal conflict [8], the current studies tracked prioritizing behaviorally during actual goal pursuit to investigate (i) if goal conflict is actually solved by prioritizing, and (ii) if older adults prioritize more than younger age groups.

The contribution of this research to the motivational and adult developmental literature is: (1) it goes *beyond* the self-report results that dominate these research areas, (2) it tests the processes of prioritization *during* goal pursuit, (3) it uses *experimental goals* instead of idiosyncratic personal goals that differ on various dimensions (e.g., time-frame, difficulty).

## Author Contributions

**Conceptualization:** Alexandra M. Freund.

**Data curation:** Martin J. Tomasik.

**Formal analysis:** Martin J. Tomasik.

**Funding acquisition:** Alexandra M. Freund.

**Methodology:** Alexandra M. Freund, Martin J. Tomasik.

**Project administration:** Alexandra M. Freund.

**Supervision:** Alexandra M. Freund.

**Writing – original draft:** Alexandra M. Freund.

**Writing – review & editing:** Martin J. Tomasik.

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
