## [Decision Letter · Decision Letter 0]

10 Nov 2020

PONE-D-20-25253

Managing Multiple Goals Through Prioritization? The Role of Age and Relative Goal Importance

PLOS ONE

Dear Dr. Freund,

Thank you for submitting your manuscript to PLOS ONE. After careful consideration, we feel that it has merit but does not fully meet PLOS ONE’s publication criteria as it currently stands. Therefore, we invite you to submit a revised version of the manuscript that addresses the points raised during the review process.

We look forward to receiving your revised manuscript.

Kind regards,

Laura Zamarian

Academic Editor

PLOS ONE

Journal Requirements:

'Review board of the Dept. of Psychology, University of Zurich; President: Prof. Dr. Klaus Oberauer, email: k.oberauer@psychologie.uzh.ch;

- no approval number was provided as the studies followed the standard protocol of the ethics committee

- written informed consent was obtained in all three studies'

a. Please amend your current ethics statement to confirm that your named institutional review board or ethics committee specifically approved this study.

4. Please ensure that you refer to Figure 4 in your text as, if accepted, production will need this reference to link the reader to the figure.

Reviewers' comments:

Reviewer's Responses to Questions

**Comments to the Author**

1. Is the manuscript technically sound, and do the data support the conclusions?

Reviewer #1: No

Reviewer #2: Yes

2. Has the statistical analysis been performed appropriately and rigorously? 

Reviewer #1: No

Reviewer #2: Yes

3. Have the authors made all data underlying the findings in their manuscript fully available?

Reviewer #1: No

Reviewer #2: No

4. Is the manuscript presented in an intelligible fashion and written in standard English?

Reviewer #1: Yes

Reviewer #2: Yes

5. Review Comments to the Author

Reviewer #1: The authors tested the associations between goal prioritization and participant age, goal conflict, and affect. Below are my comments:

1. The following sentence in the abstract "Thus, when encountering conflicting goals older adults are better in prioritization than younger adults" does not fully match the data; older adults were found to have a preference for prioritizing, but there is no evidence to suggest that they were better at it (in fact, Study 1 showed better overall performance for younger participants).

2. I would like to read more about why the authors chose their analyses and perhaps see a figure of their proposed analyses to help me understand their choice. I would strongly advocate for a paragraph and supporting figures under a separate "Data analytic approach" section before the actual analyses. For example, it is unclear to me if the authors did or did model time in their analyses; in some places, it sounds like they did not, while in others, it sounds like they might have (e.g., in the affective analyses and in their discussion of Study 1, where they write things such as "Over the course of goal pursuit").

In general, I do not understand why the authors analyzed so many different models. One of the main advantages of SEM is the ability to create larger models and analyze multiple variables at the same time. It seems odd to analyze affective experience separately from, say, perceived goal conflict. It is possible that some of the relationships the authors found may diminish in a more parsimonious model that is more effective at preventing Type I error than all these separate models.

Most importantly of all, how can we be sure that participants' age was associated with increased prioritization but more conflict and negative affect if age was not even considered in the analyses involving conflict and negative affect? This was one of the main findings highlighted by the authors, yet not directly tested. What is needed here is a single and clear model that will allow the authors to test their speculation that "Hence, when prioritizing one goal over the other despite having been instructed to solve both tasks, participants might have felt that the very act of prioritizing lead to higher conflict with the relatively neglected goal, which might have also lead to more negative affect and to feeling less in control. "

3. "However, the opposite pattern emerged: The more participants prioritized one goal over the other, the more conflict they experienced over the course of goal pursuit, the less positive their affective experience, and the less they felt in control. This is a perplexing result [...]"

I don't find this all that perplexing; participants saw both goals at all times and because they were continuously reminded of both goals, goal activation of both should have been high. It sounds from your interpretation of the findings as well as your introduction that you had expected that prioritization may have led to goal shielding, which then would have led to positive rather than negative affective experiences and perceptions. However, by literally showing participants both tasks at the same time, you effectively prevented goal shielding from happening; because of your methods, participants were likely constantly reminded about their lack of progress on the task that they were not completing. Indeed, Shah et al. (2002) found a negative association between goal shielding and negative affect, suggesting that something else happened with your participants.

Because of all the concerns cited above, I am not entirely convinced about the contributions of these findings, at least in the way that they are framed now. As per my point 3), this is a test of prioritization amidst very particular circumstances. This type of goal conflict does not seem especially ecologically valid, because in real life, we rarely have 2 or more of our conflicting goals in front of us at all times. If we did, goal shielding would be virtually impossible and perceptions of goal conflict would be on a constant high.

To see this work published, I would either like to see additional work where tasks are not presented simultaneously (e.g., allow participants to switch between them using separate tabs) or an acknowledgement of the very particular situations that your findings apply to, along with clearly defined and parsimonious analysis models. At this point, I am afraid that I cannot recommend this work for publication.

P.S. This is a minor point, but I would recommend proofreading the manuscript one final time, as I found several typos (e.g., "We had selected both tasks with by three criteria" (p. 8) and using dashes instead of em-dashes (e.g., "We hoped to achieve this by selecting one task that queries general knowledge and one that taps into verbal abilities – both skills that show little age-related decline within the studied age-range" instead of the correct "We hoped to achieve this by selecting one task that queries general knowledge and one that taps into verbal abilities—both skills that show little age-related decline within the studied age-range")).

Reviewer #2: Dear colleagues,

Thank you for the opportunity to review this manuscript for PLOS ONE. I read a very well-written paper on prioritization which is often assumed to help successfully managing multiple goals. By three studies, two with classical experimental design and a third with a reflective component thought to better explain the findings on the first two experimental studies, the authors demonstrated that if goals are equally important, prioritization still results in perceived goal conflict and is associated with negative affect, however. But: this is only the case when the equal importance of tasks is explicitly specified in the task instructions, i.e., when goal conflict does not result from an individual evaluation of the goals importance but from an externally provided interpretation of the task structure. The presentation of the studies’ methods and procedures was were clear, detailed, and in most parts comprehensible. I particularly liked the supplementary reflective study 3 with the purpose of better understanding the mechanisms behind the pattern of results in studies 1 and 2.

I have the following comments/suggestions for the manuscript:

1) When presenting the three studies and their objectives in “The present studies” (p. 6/7) could you please put some more information on the study 3 already here in the manuscript: Please state why it was conducted (to substantiate so fare speculative mechanisms suggested by the results of studies 1 and 2). Please also mention that it was an imaginary study using vignettes of study scenarios of studies 1 and 2 and what self-reports were about. This might help to give and hold track. First, when reading your manuscript, I expected study 3 to entail some kind of a thinking aloud or commenting associations/interpretations on the instructions of studies 1 and studies 2 provided to the participants.

2) For study 3, for the vignette mimicking of study 2 in the instruction it was stated that “The instruction does not specify the relative importance of the goals”. I am not sure whether it is helpful to make explicit (noticing that there is NO instruction on the relative importance of the goals) what is part of the implicit interpretation of the instructions under natural conditions, when being a participant in the experiment. I feel that it would be more important to know how the instruction is interpreted without explicitly pointing this out to the participants. Without knowing the procedures and instructions from study 1, they might not expect an additional statement on (non-specified) goal importance here. And was the term “importance of the goals” used in study 3? Was considered to describe this in everyday language? These could be points to be addressed in the discussion.

3) A power analysis was run and is reported on (p. 7). However, the authors had problems in recruiting enough participants to reach the specified sample sizes. Could the authors please explain based on what previous evidence they assumed an effect size of f = .25?

4) Throughout the manuscript exact test statistics (e.g. t-scores etc.) instead of pure p-values should be reported. Please also provide exact p-values if above p<.001 instead of p<.01 or p<.05.

5) What might be the reason for middle-aged participants having reported the lowest satisfaction with life in study 1 (p. 7). This results is also in contrast to study 3 where scores were generally substantially lower than in study 1 and study 2 (p. 23); in study 3 older adults reported the highest satisfaction with life (M = 4.42), although being on the same level as (less) satisfied younger adults in study 1 (M = 4.52) and study 2 (M = 4.54), with older adults having substantially higher scores, M = 5.11 in study 2 (l. 361) and M = 4.88 in study 1 (l. 152).

6) The authors refer to a pilot study and mention that they adjusted task difficulty to different age-groups. Please be specific about the exact adjustments made to the task (p. 8): Which no. of items had to be sorted? What does the ‘extent to which letters were scrambled’ mean specifically? Given the large heterogeneity in performance in old age, it might be critical to do a universal adjustment. Could the authors please explain their reasoning and their exact procedures a little more.

7) There are some linguistic inaccuracies in the report, e.g., l. 125 “performance based” must be “time based”; l. 171: two prepositions “with by”; l. 604/5: “the results of Study 2 […] supports (sg. instead of pl.); l. 661: preposition missing in the sentences starting “Both goals were…”; last page of general discussion seems to lack any line breaks: maybe a line break before “This is one of the general challenges…” (l. 668). I think the paper should undergo a final round of careful editing.

I hope my comments are helpful for both the editorial decision as well as to the authors.

Kind regards, Verena Klusmann (opting for signed peer review)

6. PLOS authors have the option to publish the peer review history of their article (what does this mean?). If published, this will include your full peer review and any attached files.

Reviewer #1: No

Reviewer #2: **Yes: **Verena Klusmann

---

## [Author Response · Author response to Decision Letter 0]

18 Jan 2021

Rebuttal letter

Thank you for giving us the opportunity to revise and resubmit the manuscript "Managing Conflicting Goals Through Prioritization? The Role of Age and Relative Goal Importance." We very much appreciate the time and effort the two reviewers have spent on commenting on the previous version of the manuscript. We have revised the manuscript addressing the issues raised by the reviewers, and believe that we have significantly improved the manuscript with this help. The changes are marked with “Track Changes” in the manuscript.

We apologize for the delay in submitting the revision, but we - as most people - had to juggle the unusual demands of this unusual time of the pandemic. 

Below we address each of the points raised by the reviewers point by point:

General issues:

2. a. Please amend your current ethics statement to confirm that your named institutional review board or ethics committee specifically approved this study.

Response:

The research was carried out in accordance with the regulations of the Ethics Committee of the Faculty of Arts and Sciences at the University of Zurich. These regulations define a two-stage process of ethical clearance, the first stage of which is a self-assessment of ethical risks by the researcher according to a checklist provided by the Ethics Committee. The present research passed the first stage, and therefore was exempt from further review by the Ethics Committee.

Response: The data is now publicly available at DOI 10.17605/OSF.IO/6PKBV.

4. Please ensure that you refer to Figure 4 in your text as, if accepted, production will need this reference to link the reader to the figure.

Response: Done. 

Specific Issues raised by the reviewers:

Reviewer 1:

1. The following sentence in the abstract "Thus, when encountering conflicting goals older adults are better in prioritization than younger adults" does not fully match the data; older adults were found to have a preference for prioritizing, but there is no evidence to suggest that they were better at it (in fact, Study 1 showed better overall performance for younger participants).

Response: We have changed the sentence in the Abstract accordingly: "Thus, when encountering conflicting goals older adults prioritize more than younger adults, but prioritization might not be optimal for solving short-term goal conflict when both conflicting goals are equally important."

2. I would like to read more about why the authors chose their analyses and perhaps see a figure of their proposed analyses to help me understand their choice. I would strongly advocate for a paragraph and supporting figures under a separate "Data analytic approach" section before the actual analyses. For example, it is unclear to me if the authors did or did model time in their analyses; in some places, it sounds like they did not, while in others, it sounds like they might have (e.g., in the affective analyses and in their discussion of Study 1, where they write things such as "Over the course of goal pursuit").

In general, I do not understand why the authors analyzed so many different models. One of the main advantages of SEM is the ability to create larger models and analyze multiple variables at the same time. It seems odd to analyze affective experience separately from, say, perceived goal conflict. It is possible that some of the relationships the authors found may diminish in a more parsimonious model that is more effective at preventing Type I error than all these separate models.

Response: We have included a paragraph on the Data-analytic approach clarifying the analyses we conducted: 

“Data-analytic approach. In order to account for the multiple measurements across the five rounds of the experiment, we modelled both the exogeneous (i.e., prioritization) and the endogenous (i.e., perceived task conflict and affective experience) as latent growth models including an intercept and, whenever possible, also a slope of the five subsequent measurements. Age group effects were determined by regressing the intercept and/or slope of the variable under investigation on the grouping variable. We refrained from computing multi-group models (that would have allowed testing additional hypotheses) due to the moderate size of our samples“

Concerning the specific issue of analyzing the data in full vs. split models, we have three remarks to add. First, we would agree with the Reviewer if there were several exogenous variables and only one endogenous variable. In this case, the partial regression weights could be calculated simultaneously, statistically controlling each exogeneous variable for the others, and reducing Type I error. However, in our models, we do not have several exogeneous variables and only one endogenous. The opposite is the case, as there is only one exogeneous variable (e.g., indicator of prioritization) and several endogenous variables (e.g., three dimensions of affect). In this case, adding additional endogenous variables would not change the results at all. Second, our sample sizes are just sufficient for running the analyses focusing only on one or on very few (endogenous) variables. If we had much larger sample sizes, we could have considered all endogenous variables at the same time in a large model. This, however, would not at all affect the results as the different endogenous variables would not affect each other. Finally, combining different exogenous variables (i.e., in our case, two different measures of prioritization that eventually are highly correlated) would have always been possible mathematically but we only used this modelling strategy whenever it was feasible computationally. 

Most importantly of all, how can we be sure that participants' age was associated with increased prioritization but more conflict and negative affect if age was not even considered in the analyses involving conflict and negative affect? This was one of the main findings highlighted by the authors, yet not directly tested. What is needed here is a single and clear model that will allow the authors to test their speculation that "Hence, when prioritizing one goal over the other despite having been instructed to solve both tasks, participants might have felt that the very act of prioritizing lead to higher conflict with the relatively neglected goal, which might have also lead to more negative affect and to feeling less in control. "

Response: As we did not assume that age would moderate the association between prioritization and perceived conflict, we did not test this hypothesis and planned our sample size accordingly. Also, we do not see any reason to assume such a moderation effect. Our assumption was that, irrespective of age, more prioritization would be associated with more perceived conflict. However, we hypothesized that older people would prioritize more and this is what we have found and also reported. 

3. "However, the opposite pattern emerged: The more participants prioritized one goal over the other, the more conflict they experienced over the course of goal pursuit, the less positive their affective experience, and the less they felt in control. This is a perplexing result [...]"

I don't find this all that perplexing; participants saw both goals at all times and because they were continuously reminded of both goals, goal activation of both should have been high. It sounds from your interpretation of the findings as well as your introduction that you had expected that prioritization may have led to goal shielding, which then would have led to positive rather than negative affective experiences and perceptions. However, by literally showing participants both tasks at the same time, you effectively prevented goal shielding from happening; because of your methods, participants were likely constantly reminded about their lack of progress on the task that they were not completing. Indeed, Shah et al. (2002) found a negative association between goal shielding and negative affect, suggesting that something else happened with your participants.

Response: Yes, in hindsight, we feel the same way. When having to pursue two conflicting goals, the conflict becomes even more pronounced when prioritizing one, and thereby neglecting the other. Please note, however, that the lifespan models discussed in this paper all suggest that disengaging from a conflicting goal is the key for successfully solving goal conflicts. Thus, at first glance this is a perplexing result which is why we were compelled to rerun the experiment with a different instruction and a self-report study following it up.

We agree with the reviewer that participants were likely unable to shield their goals against the competing one in the first experiment. Theoretically, when being able to shield a goal successfully (the Shah et al. findings are less convincing as one might wish given that the effect of goal shielding only emerged in interaction with regulatory focus), participants should no longer feel any conflict between the goals and also experience less negative affect as a result. This was exactly what we had originally hypothesized. Our interpretation, as laid out in the paper, is that the instruction to pursue both goals kept participants from being able to disengage from the non-prioritized goal. (we state in the discussion "Prioritization during the pursuit of two simultaneously presented tasks might not have reduced the perceived conflict because this specific set-up of both tasks on one computer screen reminds people throughout working on one of the tasks that they are doing so at the cost of the other, neglected task. This might have prevented goal shielding, a mechanism enabling temporary prioritization of the currently pursued goal (Shah et al, 2002)."

Because of all the concerns cited above, I am not entirely convinced about the contributions of these findings, at least in the way that they are framed now. As per my point 3), this is a test of prioritization amidst very particular circumstances. This type of goal conflict does not seem especially ecologically valid, because in real life, we rarely have 2 or more of our conflicting goals in front of us at all times. If we did, goal shielding would be virtually impossible and perceptions of goal conflict would be on a constant high.

Response: We respectfully disagree with this evaluation of the contribution of our studies. Importantly, we did not set out to test goal shielding. For this purpose, we would have used different designs. Instead, the purpose of this research was to test for age-related differences in prioritization to solve conflicts between goals that vie for the same resource (here: time). To make this more prominent, we have changed the title of the manuscript from “Managing Multiple Goals….” to “Managing Conflicting Goals …” Moreover, we have stressed the focus on goal conflict more clearly throughout the introduction. Thus, our research question was not how people shield their goals they pursue sequentially (which does not constitute a goal conflict in the usual definition of this term in the goal literature). The induction of a resource-based goal conflict requires the simultaneous presentation of the tasks. Else, we would have induced sequential working on tasks per design which would not allowed us to operationalize prioritization in terms of how much participants focus on one or the other task. We stress the specific boundary conditions in several places in the introduction as well as in the discussion. 

To see this work published, I would either like to see additional work where tasks are not presented simultaneously (e.g., allow participants to switch between them using separate tabs) or an acknowledgement of the very particular situations that your findings apply to, along with clearly defined and parsimonious analysis models. At this point, I am afraid that I cannot recommend this work for publication.

Response: We regret that the reviewer was not convinced that the paper makes a contribution to the literature on prioritization and goal conflicts with a particular focus on age-related differences. This is the first set of experiments investigating the hypotheses. We already followed up the first experiment with a second one that took us about one year to finish. Note, that these kinds of studies involving age-heterogeneous non-student samples of participants who have to come to the laboratory are extremely time-consuming. Adding studies is, of course, easier said than done, particularly in a time of a pandemic when in-person testing is impossible. More to the point of the contribution of the paper to the literature: we believe that the results are important for other researchers interested in how people solve goal conflicts. 

P.S. This is a minor point, but I would recommend proofreading the manuscript one final time, as I found several typos (e.g., "We had selected both tasks with by three criteria" (p. 8) and using dashes instead of em-dashes (e.g., "We hoped to achieve this by selecting one task that queries general knowledge and one that taps into verbal abilities – both skills that show little age-related decline within the studied age-range" instead of the correct "We hoped to achieve this by selecting one task that queries general knowledge and one that taps into verbal abilities—both skills that show little age-related decline within the studied age-range")).

Response: We apologize for such mistakes as using a dash instead of an em-dash (we only found the one instance mentioned by the reviewer in the manuscript) and the like. We have once again very carefully proof-read the manuscript. 

Reviewer #2: 

1) When presenting the three studies and their objectives in “The present studies” (p. 6/7) could you please put some more information on the study 3 already here in the manuscript: Please state why it was conducted (to substantiate so fare speculative mechanisms suggested by the results of studies 1 and 2). Please also mention that it was an imaginary study using vignettes of study scenarios of studies 1 and 2 and what self-reports were about. This might help to give and hold track. First, when reading your manuscript, I expected study 3 to entail some kind of a thinking aloud or commenting associations/interpretations on the instructions of studies 1 and studies 2 provided to the participants.

Response. Thank you for this suggestion. We have done so. In the abstract it now reads: "Study 3 (N = 721 young, middle-aged, older adults) was an online study using hypothetical scenarios. This study was run to substantiate the potential mechanism underlying the differences between Study 1 and 2 and supported the hypothesized effect of the instructional strictness of pursuing two goals."

In the section "present studies" we have now added: Study 3 used self-report to test these two situations of solving goal conflict through prioritization (importance to solve both tasks more or less stressed) against each other. This experiment used hypothetical scenarios describing the instructions used in the first two experiments to test if the instructional strictness might have contributed to the hypothesized differences in results between Study 1 and 2.

2) For study 3, for the vignette mimicking of study 2 in the instruction it was stated that “The instruction does not specify the relative importance of the goals”. I am not sure whether it is helpful to make explicit (noticing that there is NO instruction on the relative importance of the goals) what is part of the implicit interpretation of the instructions under natural conditions, when being a participant in the experiment. I feel that it would be more important to know how the instruction is interpreted without explicitly pointing this out to the participants. Without knowing the procedures and instructions from study 1, they might not expect an additional statement on (non-specified) goal importance here. And was the term “importance of the goals” used in study 3? Was considered to describe this in everyday language? These could be points to be addressed in the discussion.

Response: We double-checked the instruction with two native English speakers who did not raise any concerns about the language. We have now asked two more native speakers, and they all thought that the term "importance of goals" is very much part of everyday language. Given that we cannot go back to the participants of Study 2, unfortunately we cannot probe qualitatively how they interpreted the instruction. We agree that this would have been a worthwhile endeavor to substantiate our interpretation of this study. Regarding the instruction in Study 3: we were worried that not including that no mentioning of the relative importance of the goals was made, participants would infer that the experimental situation would demand people to pursue both goals with equal intensity. Obviously, this is an empirical question and we are unable to answer it with the data of this study. 

3) A power analysis was run and is reported on (p. 7). However, the authors had problems in recruiting enough participants to reach the specified sample sizes. Could the authors please explain based on what previous evidence they assumed an effect size of f = .25?

Response: This effect size was based on the self-report study by Riediger and Freund (2006).

4) Throughout the manuscript exact test statistics (e.g. t-scores etc.) instead of pure p-values should be reported. Please also provide exact p-values if above p<.001 instead of p<.01 or p<.05.

Response: We now report the exact test statistics and the exact p-values according to the rules of the 7th edition of the APA Citation Style Manual.

5) What might be the reason for middle-aged participants having reported the lowest satisfaction with life in study 1 (p. 7). This results is also in contrast to study 3 where scores were generally substantially lower than in study 1 and study 2 (p. 23); in study 3 older adults reported the highest satisfaction with life (M = 4.42), although being on the same level as (less) satisfied younger adults in study 1 (M = 4.52) and study 2 (M = 4.54), with older adults having substantially higher scores, M = 5.11 in study 2 (l. 361) and M = 4.88 in study 1 (l. 152).

Response: There are a number of studies showing that life-satisfaction might be lowest in the middle-aged group (e.g., Cheng, Powdthavee, & Oswald, 2017; Blanchflower & Oswald, 2017; 2019) while other researchers argue that this is not the case (e.g., Galambos, Fang, Krahn, Johnson, & Lachman, 2015). There is certainly some variation between measures of life satisfaction as well as between samples. As life-satisfaction was not in any way a focus of this set of studies, we would like to not extend the discussion to this variation in this measure. However, we have now mentioned it in the discussion: “Note, however, that the samples of the three studies were not entirely equivalent in terms of some of the background variables (e.g., the association of age and the single-item measure of life-satisfaction). Such sample variations in measures of imperfect reliability are to be expected. 

6) The authors refer to a pilot study and mention that they adjusted task difficulty to different age-groups. Please be specific about the exact adjustments made to the task (p. 8): Which no. of items had to be sorted? What does the ‘extent to which letters were scrambled’ mean specifically? Given the large heterogeneity in performance in old age, it might be critical to do a universal adjustment. Could the authors please explain their reasoning and their exact procedures a little more.

Response: Task difficulty of the item sorting task was operationalized as the number of items to be sorted and for the word riddle task as the minimum number of letter swaps necessary to solve the riddle. The latter operationalization is reminiscent of the Levenshteyn distance used in information theory or linguistics. In the pilot study, we presented participants with different levels of difficulty and decided for those in which the different age groups showed a similar performance. We used 9, 11, and 12 items for the three age groups, respectively.

7) There are some linguistic inaccuracies in the report, e.g., l. 125 “performance based” must be “time based”; l. 171: two prepositions “with by”; l. 604/5: “the results of Study 2 […] supports (sg. instead of pl.); l. 661: preposition missing in the sentences starting “Both goals were…”; last page of general discussion seems to lack any line breaks: maybe a line break before “This is one of the general challenges…” (l. 668). I think the paper should undergo a final round of careful editing.

Response: Thank your pointing us to these mistakes for which we apologize. We have corrected them.

I hope my comments are helpful for both the editorial decision as well as to the authors.

Response: Yes, thank you very much for your constructive and helpful comments.

We appreciate the issues both of the reviewers raised and hope that we have been able to address them in a satisfactory way in this revision.

---

## [Editor Report · Decision Letter 1]

1 Feb 2021

Managing Goal Conflict Through Prioritization? The Role of Age and Relative Goal Importance

PONE-D-20-25253R1

Dear Dr. Freund,

We’re pleased to inform you that your manuscript has been judged scientifically suitable for publication and will be formally accepted for publication once it meets all outstanding technical requirements. 

Kind regards,

Laura Zamarian

Academic Editor

PLOS ONE

Additional Editor Comments (optional):

Please check again the manuscript. There are still some typos (e.g., when describing the study by Ballard at al. the first time a "not" is missing; the number of estimated participants in Study 3 should be "169" and not "69"). Some verbs are missing or not correct.
---

## [Editor Report · Acceptance letter]

8 Feb 2021

PONE-D-20-25253R1 

Managing conflicting goals through prioritization? The role of age and relative goal importance 

Dear Dr. Freund:

I'm pleased to inform you that your manuscript has been deemed suitable for publication in PLOS ONE. Congratulations! Your manuscript is now with our production department. 

Kind regards, 

on behalf of

Dr. Laura Zamarian 

Academic Editor

PLOS ONE